# Fatty Acid Metabolism Regulators Have Pivotal Roles in the Pathogenesis of Ovarian Carcinoma

**DOI:** 10.3390/ijms26104794

**Published:** 2025-05-16

**Authors:** Megumi Watanabe, Motoki Matsuura, Tatsuya Sato, Makoto Usami, Tsuyoshi Saito, Masato Furuhashi, Kohichi Takada, Hiroshi Ohguro

**Affiliations:** 1Department of Ophthalmology, School of Medicine, Sapporo Medical University, S1W17, Chuo-ku, Sapporo 060-8556, Japan; watanabe@sapmed.ac.jp; 2Department of Obstetrics and Gynecology, Sapporo Medical University, S1W17, Chuo-ku, Sapporo 060-8556, Japan; motoki.gyne@gmail.com (M.M.); desmoplakin@gmail.com (T.S.); 3Department of Cardiovascular, Renal and Metabolic Medicine, Sapporo Medical University, S1W17, Chuo-ku, Sapporo 060-8556, Japan; satatsu.bear@gmail.com (T.S.); furuhasi@sapmed.ac.jp (M.F.); 4Department of Cellular Physiology and Signal Transduction, Sapporo Medical University, S1W17, Chuo-ku, Sapporo 060-8556, Japan; 5Department of Medical Oncology, Sapporo Medical University, S1W17, Chuo-ku, Sapporo 060-8556, Japan

**Keywords:** ovarian carcinoma, a pan-cancer analysis, Seahorse Bioanalyzer, spheroid

## Abstract

To study the pathological contribution of fatty acid (FA) metabolism regulators including fatty acid binding protein 4 (FABP4), FABP5, peroxisome proliferator-activated receptor alpha (PPARα), and PPARγ in ovarian carcinoma, non-cancerous human ovarian surface epithelium (HOSE) cells and two epithelial ovarian carcinoma (EOC) cell lines, AMOC-2 and ES2 established from ovarian serous adenocarcinoma and ovarian clear cell carcinoma, respectively, were subjected to (1) an analysis of the physical properties of spheroids, (2) qPCR analysis, (3) cellular metabolic analysis, and (4) multiomic pan-cancer analysis using the Cancer Genome Atlas (TCGA). In contrast to globe-shaped spheroids of HOSE cells, AMOC-2 and ES2 cells formed non-globe-shaped spheroids and ES2 spheroids were much more fragile than AMOC-2 spheroids. Gene expression levels of FABP4 and FABP5 in AMOC-2 cells and those of PPARγ in AMOC-2 cells were significantly higher than those in HOSE cells. Metabolic phenotypes and the effectiveness against antagonists for regulators were significantly different in the two types of cancerous cells. Those regulators were identified by a multiomic pan-cancer analysis as novel factors for the prediction of the prognosis of ovarian serous adenocarcinoma. The results show that dysregulated FA metabolism in AMOC-2 and ES2 suggests that the regulation of FA metabolism may be a critical factor in the pathogenesis of EOC.

## 1. Introduction

Epidemiological studies have revealed that ovarian carcinoma (OC) is the eighth most common carcinoma in women [1] and the fifth major cause of carcinoma-related deaths in women over 40 years of age [2]. Pathologically, OC is a heterogeneous malignancy that consists of epithelial OC (EOC) and non-epithelial OC (NEOC). EOC is the predominant OC and is further categorized into high-grade serous carcinoma (HGSC), the most common type that accounts for 60–70% of all cases, ovarian clear cell carcinoma (OCCC), the second most common OC that accounts for 5–24% of EOC cases, and other rare types of EOC including endometroid OC and mucinous OC [3,4,5,6,7]. Although these EOC subtypes have distinct cellular origins and different genetic and molecular aspects, taxane- and platinum-based anticancer drugs are used as a standard regimen of chemotherapy regardless of the EOC subtype [8,9]. Since an individualized regimen of chemotherapy for each EOC subtype would be needed for good treatment outcomes, additional therapeutic targets that can inhibit the progression of or even cure each EOC subtype will be required.

Premature menstruation, nulligravida, advanced maternal age, and late menopause have been shown to be risk factors for EOC progression [10]. Obesity is also recognized as a risk factor for the development and progression of EOC and other malignant tumors [11,12,13]. It has been shown that obesity is negatively correlated with the survival of patients with EOC by affecting their metastasis [14], and a high visceral fat-to-muscle ratio is known as an independent factor for predicting deteriorated overall survival (OS) in patients with EOC [15]. These findings collectively suggest that fatty acid (FA) metabolism factors such as adipokines may be involved in the pathogenesis of EOC, and, in fact, within tumor surrounding microenvironments, which have crucial roles in tumorigenesis tumor progression, and metastasis [16,17,18,19], and several adipokines in addition to other factors including monocyte chemotactic protein-1 (MCP-1), interleukin 6 (IL-6), interleukin 8 (IL-8), tissue inhibitor of metalloproteinase-1 (TIMP-1), vascular endothelial growth factor-A (VEGF-A), and matrix metalloproteoinase-2 (MMP-2) have been detected [20,21,22]. In fact, among various FA metabolism regulators, previous studies have suggested that fatty binding acid protein 4 (FABP4) [23,24] and peroxisome proliferator-activated receptors (PPARs) [25,26,27] are expressed in the ovarian tissues and are involved in the pathogenesis of OC. In addition, it was shown that FABP5 was also identified as a pivotal factor regulating the pathogenesis of various malignant tumors [28]. However, at the time of this writing, the contribution of these FA metabolism regulators to the pathogenesis of EOC has not been intensively investigated.

Recently, in vitro models using three-dimensional (3D) culture systems have been increasingly used in the tumor research field in addition to the conventional two-dimensional (2D) cell culture method [29,30]. Various methodologies have been developed to produce single cell-derived spheroids from cancerous cells for the evaluation of drug toxic effects and for the discovery of new anti-cancer drugs [29,30]. In our recent studies, we have established various in vitro tumor spheroid models using malignant melanoma cell lines [31] and oral squamous cell carcinoma cell lines [32] by a hanging drop culture method, and we found non-globe-shaped configurations in these tumor spheroids that were totally different from globe-shaped spheroids generated from non-cancerous cells [31,32,33,34]. Furthermore, the appearances of spheroids generated from malignant tumor cells were diverse, and the differences in appearances were correlated with some biological functions such as essential cellular metabolic functions assessed by a Seahorse Bioanalyzer and sensitivities to anti-tumor drugs [31]. These findings suggested that the appearance of 3D tumor spheroids may be a key index of some important biological aspects of malignant tumors. Collectively, it is of great interest to know the possible contribution of FA metabolism regulation to the pathogenesis of EOC and the biological aspects of 3D EOC spheroids.

To study those issues, in addition to non-cancerous human ovary surface epithelium (HOCE) cells, two EOC cell lines, AMOC-2 cells established from ovarian serous adenocarcinoma (OSC), and ES2 cells established from the OCCC, which are the two most common types of EOC in Japan [7], were subjected to the following analyses: (1) the analysis of spheroid configuration, (2) qPCR analysis for cell junction molecules including ZO1 and claudin1 and for possible candidates of FA metabolism regulators including FABP4, FABP5, PPARα, and PPARγ, (3) multiomic pan-cancer analysis using the Cancer Genome Atlas (TCGA) [35,36], and (4) Seahorse cellular metabolic analysis using specific inhibitors for these FA metabolism factors.

## 2. Results

In previous studies, it was shown that non-globe-shaped tumor cell-derived spheroids were produced by using a hanging drop culture method, in which only the gravity force and buoyant force are required for generating spheroids [31,32,33,34,37,38,39,40,41], suggesting that the appearance of spheroids may be a key index for some important biological aspects of tumorgenesis [42]. Initially, to compare the configuration of spheroids generated from non-cancerous HOSE cells, two EOC cell lines, AMOC-2 cells established from OSC and ES2 cells established from OCCC cells, downward and lateral images of spheroids were observed by a phase contrast microscope and a microscope monitoring camera, respectively, as described in our recent report [43]. As shown in Figure 1, in contrast to the globe-shaped configuration of non-cancerous HOSE spheroids, (1) ES2 cell-derived spheroids in the downward images were circular and AMOC-2 cell-derived spheroids in the downward images were also circular with some deformity, and (2) the lateral configuration of AMOC-2 cell-derived spheroids was a disc-shape, but the lateral configuration of ES2 cell-derived spheroids could not be observed because ES2 cell-derived spheroids easily came apart during their transfer to the microplate. Furthermore, the size of non-cancerous HOSE spheroids was significantly smaller than that of cancerous spheroids, in which ES2 spheroids were much larger than AMOC-2 spheroids, suggesting that cell aggregation to form a spheroid of cancerous cells was substantially weaker than that of non-cancerous HOCE cells and such weak cell aggregation in cancerous cells was more evident in ES2 cells. As the possible underlying mechanisms causing larger and fragile spheroid formation caused by weak cell aggregation in OC cell lines, it was speculated that cell junction properties and lipid-related factors may be involved. To test this hypothesis, levels of expression of cell junction molecules including ZO-1 and claudin1, and lipid metabolism regulators including FABP4, FABP5, PPARα, and PPARγ were compared among non-cancerous HOCE and cancerous AMOC-2 and ES2.

In terms of the gene expression of cell junction molecules, the levels of mRNA expression of ZO1 and claudin1 in AMOC-2 cells and the levels of mRNA expression of claudin1 in ES2 cells were significantly lower than those in HOSE cells (Figure 2). These observations supported the results showing that cancerous AMOC-2 spheroids and ES2 spheroids were more fragile than non-cancerous HOCE spheroids. Alternatively, qPCR analysis and immunocytochemistry showed that the expression of FABP4 and FABP5 were significantly higher in ES2 cells than in HOCE cells and AMOC-2 cells, and the expression of PPARγ in AMOC-2 cells was substantially increased compared to those of HOCE cells and ES2 cells (Figure 3). Based on these results, it was speculated that FABP4 and FABP5 or PPARγ may be involved in the pathogenesis of ES2 cells or AMOC-2 cells, respectively, in addition to their fragile formation of spheroids compared to non-cancerous HOCE spheroids.

To further investigate the potential metabolic contribution of FABPs and PPARs to the biological properties of cancerous OC, Seahorse cellular metabolic analysis was performed. Given the differential expression of FABP4, FABP5, and PPARγ among HOSE, ES2, and AMOC-2 (Figure 3), we also analyzed the metabolic phenotype of ES2 and AMOC-2 following intervention with these pharmacological inhibitors as follows: a FABP4 inhibitor (10 μM BMS309430), a FABP5 inhibitor (10 μM FABP ligand 6), and a PPARγ inhibitor (0.1 μM T0070907). The metabolic characteristics of non-cancerous HOSE cells treated with inhibitors of these lipid regulators are shown in Figure 4. Although PPARγ inhibition tended to suppress cellular metabolism, there were no statistically significant differences in any of the metabolic indices in the HOSE cells. We also show overviews of the measurements of the metabolic analysis (Figure 5A,B) along with the calculated metabolic indices (Figure 5C–I) in the AMOC-2 and ES2 cells. Compared to non-cancerous HOSE cells, both AMOC-2 and ES2 exhibited a significant decrease in maximal respiration (Figure 5F). Additionally, non-mitochondrial respiration was markedly reduced in AMOC-2, whereas ES2 showed a significant increase in the baseline OCR/ECAR ratio, which reflects the ratio of oxygen consumption to glycolysis at the baseline (Figure 5I). Inhibitors targeting FABP4, FABP5, and PPARγ did not affect basal respiration or ATP-linked respiration in either AMOC-2 or ES2 (Figure 5C,D). However, these inhibitors significantly canceled the suppression of maximal respiration observed in AMOC-2 and ES2 compared to HOSE (Figure 5F). Proton leak, an index of mitochondrial uncoupling, showed a slight increase in ES2 with these lipid metabolism regulators, while the change was minimal in AMOC-2 (Figure 5E). Furthermore, these inhibitors had little effect on the spare respiratory capacity (Figure 5G) and non-mitochondrial respiration (Figure 5H), yet the significantly elevated baseline OCR/ECAR ratio in ES2 was markedly reduced by FABP5 inhibition (Figure 5I). Collectively, these findings suggest that the regulators FABP4, FABP5, and PPARγ differently affected the biological functions of AMOC-2 and ES2 cells, and the impaired mitochondrial respiratory capacity observed in ES2 and AMOC-2 may be improved by inhibiting FABP4, FABP5, and PPARγ, which strongly supports the above hypothesis that these FA metabolism regulators may be potential key pathological factors in EOC but not non-cancerous ovarian cells.

To test our hypothesis that FABP4, FABP5, and PPARγ may play pivotal roles in the pathogenesis of EOC, a possible relationship between these FA metabolism regulators and the pathogenesis of ECO, the prognostic significance of these regulators in SOC patients was studied by a Kaplan–Meier plotter using an online database. As shown in Figure 6, (1) SOC patients with a higher expression level of FABP4 or FABP5 had significantly shorter progression-free survival (PFS) and overall survival (OS) than those for SOC patients with a lower expression level, and (2) SOC patients with a higher expression level of PPARγ had significantly shorter PFS than those for SOC patients with a lower expression level. Collectively, these results suggested that the FA metabolism regulators FABP4, FABP5, and PPARγ are indeed involved in the pathogenesis of EOC and that these regulators would therefore serve as diagnostic markers or therapeutic targets for EOC.

## 3. Discussion

FAs have various important physiological roles including the maintenance of the membrane architecture, energy production by β-oxidation, and extracellular signaling [44]. It has also been shown that FAs affect the survival, proliferation, and motility of cancerous cells [45] including OC cells [46]. Recently, Kado et al. showed that oleic acid (OA) stimulates glycolysis in an HNOA ovarian adenocarcinoma cell line via PPARα, thereby regulating cancer cell growth [47]. In fact, it has been suggested that various deregulated FA metabolic pathways are involved in the pathogenesis of OC [48]. In the current study, we found quite different expression profiles of three FA metabolism regulatory factors including FABP4, FABP5, and PPARγ (Figure 3), that is, the expression of FABP4 and FABP5 and expression of PPARγ were predominant in OCCC-derived ES2 cells and OSC-derived AMOC-2 cells, respectively, suggesting that dysregulated FA metabolic mechanisms may be involved in the pathogenesis of EOC. Notably, the present extracellular flux analysis showed that the pharmacological inhibition of FABP4, FABP5, and PPARγ significantly ameliorated the reduced maximum respiration in ovarian carcinoma cell lines without affecting basal respiration (Figure 5). Although the detailed molecular mechanisms of these phenotypes remained unelucidated, it was speculated that the findings at least suggest that the pharmacological inhibition of FABP4, FABP5, and PPARγ may fine-tune mitochondrial energy metabolism presumably by modulating electron transfer efficiency in the mitochondrial electron transport chain system, the driving force of proton pumping, and/or the supply of NADH and FADH_2_ through the TCA cycle and β-oxidation in two EOC cell lines, AMOC-2 and ES2 (Figure 7).

Pathologically, OCCC originates not from the ovarian epithelium but from the transformation of displaced endometriotic tissues [49,50,51]. Interestingly, it was shown that the expression levels of the FABP4 gene were substantially elevated in the ectopic endometrium compared with that in the eutopic endometrium in women [52], in which the infiltration and activation of FABP4-expressing macrophages in ectopic endometrial sites are thought to be related [53]. A previous study revealed that BMS309403, a specific inhibitor of FABP4, induced not only a significant reduction in the metastasis of OC in a mouse model but also an increase in the sensitivity of OC cells to the carboplatin [23]. Those FABP4-related effects are in fact recognized as clinical manifestations of OCCC, that is, resistance to standard chemotherapy and a poor prognosis for advanced and recurrent OCCC [24]. It was also shown that miR-455-induced downregulation of FABP4 resulted in a significant reduction in apoptosis and oxidative stress in a human endometrial stromal cell line (HESCs) [54], suggesting that FABP4 may have important roles in the development of oxidative stress-induced endometriosis. Higher expression levels of FABP4 and vascular endothelial growth factor (VEGF) were also detected in adipose tissues adjacent to endometriotic lesions [53]. Collectively, these observations suggest that FABP4 is potentially a critical biomarker for tumor growth, metastasis, and anti-tumor drug sensitivity of OC.

PPARs are a family of transcription factors that function in diverse biological mechanisms including lipid metabolism, angiogenesis, tissue remodeling, cell cycle, and apoptosis [55]. PPARα, PPARδ, and PPARγ are expressed in the ovary and are involved in critical roles for ovarian physiology and ovarian pathology, and several drugs such as fibrates and thiazolidinediones are therefore currently in clinical use to manipulate the activities of PPARs [25,26]. In fact, PPARγ was identified as a potential therapeutic target for the suppression of tumor growth of OSC by using a murine experimental model with OSC [27]. Furthermore, a high expression level of PPARγ was identified as a predictor for poor prognosis for OSC [56] as well as a potential biomarker for predicting the recurrence of OSC based on multiomic pan-cancer analysis using TCGA [57]. These collective findings rationally support our results showing increased expression levels of PPARγ and FABP4 in the OSC cell line AMOC-2 and the OCCC cell line ES2, respectively. At the time of this writing, although the contribution of FABP5 to the pathogenesis of EOC has not been elucidated, it has been shown that FABP5 can activate various transcription factors leading to an increase in the expression of tumorigenesis-related proteins and is thus involved in the pathogenesis of various malignant tumors [28]. Therefore, this result suggests that FABP5 may also be involved in the pathogenesis of EOC, in addition to FABP4 and PPARγ. In addition, the present study showed that the pharmacological inhibition of these lipid regulatory modulators has diverse effects on mitochondrial functions, which can be associated with tumor progression and metastasis. Furthermore, a previous screening study using the liver, retina, and retinal pigment epithelium (RPE)-choroid showed that PPARα activation increased the expression of ocular FABP4 [58] suggesting that PPARα may also be involved in the pathogenesis of EOC through some unidentified linkage with FABP4. In support of this idea, the levels of PPARα expression in HOSE cells were slightly lower than those in AMOC-2 and ES2 cells, and SOC patients with a lower expression level of PPARα (Figure 3) and a Kaplan–Meier plotter using an online database showed significantly shorter PFS and overall survival (OS) than those for SOC patients with their higher expression level of PPARα (Figure 8).

In contrast to the globe-shaped appearance of spheroids generated by non-cancerous cells [31,32,33,34] including HOCE cells, the configuration of AMOC-2 spheroids was a non-globe shape as observed in other malignant tumor cell lines including the A549 cell [59], malignant melanoma cell lines [31], and oral squamous cell carcinoma cell lines [32]. ES2 spheroids were very fragile, unlike most spheroids generated from the above malignant tumor cell lines and most non-cancerous cells including human trabecular meshwork (HTM) cells [60], human conjunctival fibroblasts (HconFs) [61], and 3T3-L1 cells [62]. Such a fragile spheroid formation was also observed in various OCCC-derived cell lines generated by a static suspension culture using 24-well ultra-low attachment plates [63]. These collective observations suggested that a spheroid culture using a hanging drop culture plate is useful for the evaluation of biological aspects of EOC cell lines with very poor efficacy for cell aggregation to form spheroids compared to non-cancerous cells including HOSE cells. If that is the case, we rationally speculate that clinical manifestations of OCCC including a prompt increase in mass in a short period and short-term recurrences after treatment [51] may be explained by such poor efficacy of spheroid formation.

Nevertheless, as for the study limitations, the following issues need to be investigated. Firstly, only three distinct cell lines derived from a healthy ovarian cell line, HOSE, and two EOC cell lines, AMOC-2 and ES2, were used in the present study. Even though insufficient numbers of cell lines were used in the present study, our pilot study suggested that FA metabolism regulators may pivotally be involved in the pathogenesis of EOC based on (1) the non-globe-shaped EOC spheroid configuration (vs. a globe-shaped HOSE spheroid) and (2) greater changes in cellular metabolic functions of EOC cell lines by the pharmacological inhibition of FA metabolism regulators compared to HOSE, in addition to the prognosis estimation of OSC by analysis by a Kaplan–Meier plotter using an online database. Secondly, although cellular metabolic functions of two EOC cell lines but not non-cancerous HOSE were significantly modulated by the pharmacological suppression of FA metabolism regulators, underlying mechanisms inducing different metabolic phenotypes between AMOC-2 cells (glycolysis dominant) and ES2 cells (mitochondrial respiration dominant) assessed by a Seahorse Bioanalyzer remain to be elucidated. Since it has been shown that more than half of OCCC cases are associated with inactivating mutations of ARID1A, a component of the SWI/SNF nucleosome remodeling complex [64], and that ARID1A-deficient cells upregulate OXPHOS [65], an ARID1A mutation may be involved. However, another study showed that although OCR indices of the OCCC cell line ES2 with an ARID1A wildtype and another OCCC cell line, TOV-21G cells, with an ARID1A mutation were substantially higher than those of OACC cell lines, levels of OCR indices of ES2 cells and TOV-21G cells were comparable, suggesting that the difference in metabolic phenotypes between ES2 cells and AMOC-2 cells may not be related to an ARID1A mutation. Thirdly, possible pathophysiological roles of FABPs and/or PPARs in EOC cells have not been identified despite the fact that their specific inhibitors substantially altered cellular metabolic functions. Therefore, further studies to reveal those unidentified issues in conjugation with additional investigations using additional various EOC cell lines to find new key molecules related to FA metabolism regulators and the correlation between the expression of FA metabolism regulators and various clinical parameters, such as histological grade and FIGO staging in patients with EOC, will be our next projects.

## 4. Materials and Methods

As according to the tenets of the Declaration of Helsinki, the present study using human-derived cell lines, which was performed at Sapporo Medical University Hospital, Japan, was permitted by the Institutional Review Board (IRB registration number 342-3416).

### 4.1. Two-Dimensional and Three-Dimensional Cell Cultures of Ovarian Surface Epithelium Cell and Ovarian Carcinoma Cell Lines AMOC-2 and ES2

AMOC-2, an ovarian serous adenocarcinoma (OSC) cell line, was generously supplied by Dr. Hiromitsu Yabushita, Department of Obstetrics and Gynecology, Aichi Medical College [66]. ES2, an ovarian clear cell carcinoma (OCCC) cell line purchased from the American Type Culture Collection (ATCC, Manassas, VA, USA) was isolated from the ovary of a Black, 47-year-old, female human with clear cell carcinoma. Human ovarian surface epithelium (HOSE) cells were generously supplied by Dr. Rumi Sasaki, Department of Obstetrics and Gynecology, Kumamoto University School of Medicine [67]. In brief, ovarian surface (OSE) cells were surgically collected by scraping with a surgical blade from the ovaries that were grossly normal and no pathological lesions in subsequent histological examination were observed. Thereafter, HOSE cells were immortalized without viral oncogenes. By using a DMEM medium supplemented with 10% FBS, 1% L-glutamine, and 1% Antibiotic-Antimycotic, these cell lines were cultured in 2D planar dishes (150 mm diameter) under 5% CO_2_ at 37 °C.

The 2D and 3D cell cultures of these cell lines were processed during a period of 7 days as we reported previously [68,69]. Briefly, as for spheroid generation, a 2D cultured cell pellet was collected and resuspended in the above DMEM culture medium supplemented with methylcellulose (Methocel A4M), in which cell density was adjusted at 20,000 cells/28 μL. A 28-μL aliquot of the cell suspension was subjected in each well of a hanging drop culture plate (# HDP1385, Sigma-Aldrich, St. Louis, MO, USA). The spheroid culture was processed by daily medium changing of half of the medium in each well.

### 4.2. Analysis of Morphological Aspects of Spheroids Derived from HOSE, AMOC-2, and ES2 Cells

To study the morphological aspects of the spheroids, each spheroid was observed downwardly and laterally by using phase contrast microscopy as in our previous reports [33]. Briefly, the horizontal configurations of spheroids that were dropped into a Petri dish from wells of a 3D drop culture plate were observed by a phase contrast microscope. The spheroids were then transferred onto a microplate equipped with a MicroSquisher (CellScale, Waterloo, ON, Canada) for the observation of lateral images of spheroids observed by an obtained microscope monitoring camera. The measurement of the horizontal diameter and lateral height of 3D spheroids were processed by using Image-J software version 1.51n (National Institutes of Health, Bethesda, MD, USA) as described in our recent report [43].

### 4.3. Immunocytochemistry

As shown in our previous study, 2D cultures of HOSE, AMOC-2, and ES2 cells were fixed with 4% paraformaldehyde and blocked with 3% bovine serum albumin. Then, those were successively incubated with the 1st antibody (1:200 dilution) composed of the anti-human FABP4 rabbit antibody (ab13979, Abcam, Cambridge, UK), anti-human FABP5 rabbit antibody (30217P1120, HycultBiotech, Uden, The Netherlands), anti-human PPARα rabbit antibody (ab3484, Abcam, Cambridge, UK), or anti-human PPARγ rabbit antibody (ab59256, Abcam, Cambridge, UK) at 4 °C overnight, and the 2nd antibody mixture composed of the anti-rabbit IgG rat antibody (488 nm, 1:1000 dilution, A11072, Thermo Fischer, Waltham, MA, USA), phalloidin (594 nm, 1:1000 dilution, A12381, Thermo Fischer, Waltham, MA, USA), and DAPI (1:1000 dilution, DOJINDO, Tokyo, Japan) for 3 h. After mounting with a cover glass, immunofluorescent images were observed by a Nikon A1 confocal microscopy (Nikon Co., Tokyo, Japan).

### 4.4. Real-Time Cellular Metabolic Function Analysis of HOSE, AMOC-2, and ES2 Cell Lines

As we reported previously [31,32], the oxygen consumption rate (OCR) and extracellular acidification rate (ECAR) of the 2D cultured HOSE, AMOC-2, and ES2 cells were evaluated using a Seahorse XFe96 real-time metabolic analyzer (Agilent Technologies, Santa Clara, CA, USA) according to the manufacturer’s instructions. In particular, the AMOC-2 and ES2 cells used in this experiment were treated with FABP4 (10 μM BMS309430, Catalog #10010206, Cayman Chemical, Ann Arbor, MI, USA), FABP5 (10 μM FABP ligand 6, catalog #10010206, Cayman Chemical, Ann Arbor, MI, USA), or PPARγ (0.1 μM T0070907, catalog #S2871, Cellect Chemicals, Houston, TX, USA) or not treated (NT). The normalization of the OCR and ECAR values was calculated by the amount of protein in each well by using a BCA protein assay (TaKaRa Co., Shiga, Japan).

### 4.5. Databases and Gene Expression Data Analysis

We analyzed the correlations of the expression levels of FABP4, FABP5, PPARα, and PPARγ with the clinical outcomes of patients with serous ovarian carcinoma by using a Kaplan–Meier plotter (https://kmplot.com/analysis, accessed on 3 July 2024). The cutoff value was the “auto select best cutoff” calculated from the *p*-value and HR value [70]. In total, 1232 serous ovarian carcinoma patients were divided into two groups with high or low expression levels for each gene. Kaplan–Meier analysis of survival based on these groups was performed. The Kaplan–Meier plotter results were outputted as text and graphed using EZR software version 1.68.

### 4.6. Other Methods

As we reported previously [31,32], quantitative PCR was carried out using specific primers (Appendix A), and statistical analyses were performed by using a Graph Pad Prism 8 (GraphPad Software, San Diego, CA, USA). The statistical difference between the groups and among the groups was estimated by using the Student’s *t*-test for two-group comparison and one-way ANOVA followed by Tukey’s HSD (Honestly Significant Difference) post hoc analysis, respectively.

## 5. Conclusions

The results revealed that FA metabolism regulators including FABP4, FABP5, and PPARγ are involved in biological aspects in two EOC cell lines, AMOC-2 cells established from ovarian serous adenocarcinoma (OSC) and ES2 cells established from OCCC. In addition, an analysis by a Kaplan–Meier plotter using an online database showed that FA metabolism regulators could be novel prognostic factors in SOC patients. Collectively, the results showing that dysregulated FA metabolism may be a critical factor in the pathogenesis of EOC have prompted us to perform further studies to explore additional unidentified clinicopathological aspects of FA metabolism regulators, which will require using additional in vitro and in vivo experiments.

## Figures and Tables

**Figure 1 ijms-26-04794-f001:**
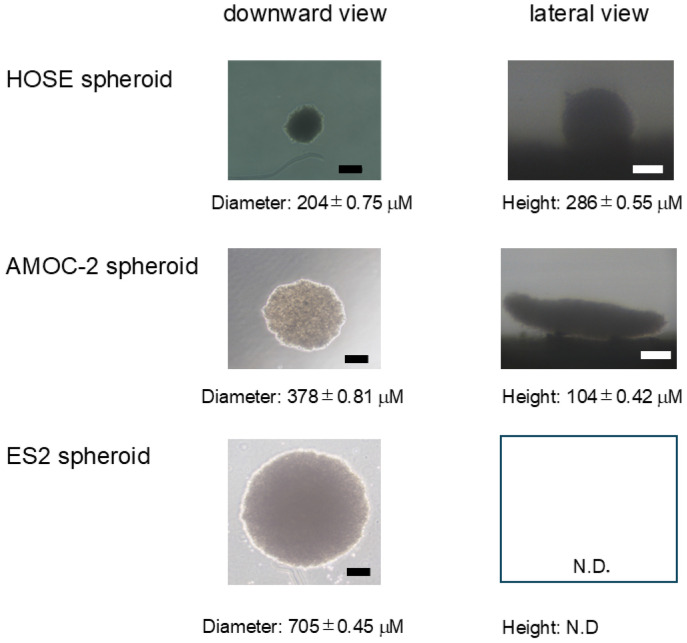
Representative phase contrast microscopy images of spheroids derived from HOSE, AMOC-2, and ES2 cells. Representative downward and lateral images of spheroids obtained from HOSE cells, AMOC-2 cells, and ES2 cells. Scale bar: 100 μm. N.D.: not determined because spheroids came apart during their transfer to the microplate.

**Figure 2 ijms-26-04794-f002:**
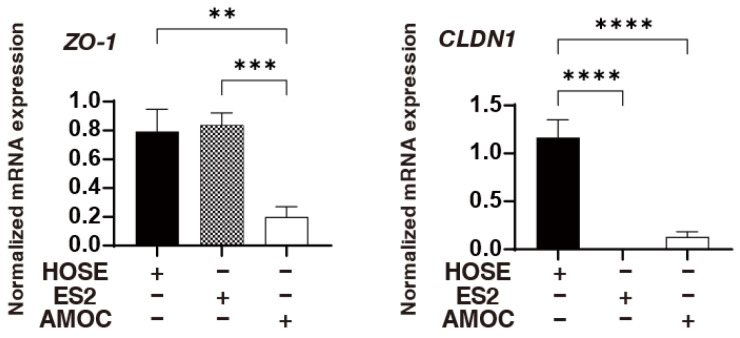
qPCR analysis of ZO1 and claudin1 in HOSE, AMOC-2, and ES2 cells. The mRNA expression levels of *ZO1* and *claudin1* in 2D cultured HOSE cells, AMOC-2 cells, and ES2 cells were evaluated by a qPCR procedure. Experiments were repeated three times with freshly prepared cells (n = 3) in each experimental condition. ** *p* < 0.01, *** *p* < 0.005, **** *p* < 0.001.

**Figure 3 ijms-26-04794-f003:**
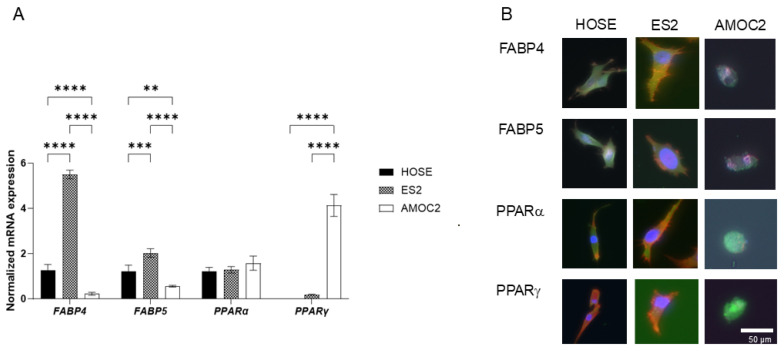
Expression of FA metabolism factors of HOSE, AMOC-2, and ES2 cells. The mRNA expression levels by a qPCR procedure (**A**) and immunostaining (**B**) of *FABP4*, *FABP5*, *PPARα*, and *PPARγ* in 2D cultured HOSE cells, AMOC-2 cells, and ES2 cells are shown. Experiments were repeated three times with freshly prepared cells (n = 3) in each experimental condition. ** *p* < 0.01, *** *p* < 0.005, **** *p* < 0.001.

**Figure 4 ijms-26-04794-f004:**
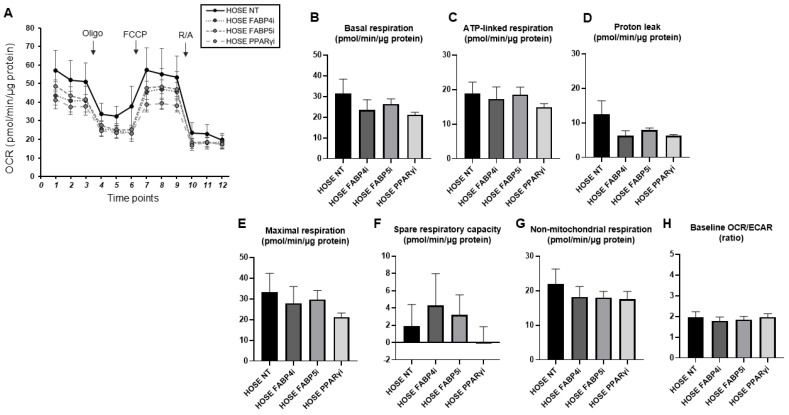
Metabolic functions of HOSE cells treated with lipid metabolism modulators. Real-time cellular metabolic function analysis of HOSE cells that were treated with inhibitors of FABP4, FABP5, and PPARγ was performed to evaluate the metabolic functions. OCR and ECAR were measured at the baseline and upon subsequent sequential supplementation with the following: Oligo, a complex V inhibitor, oligomycin; FCCP, a protonophore, carbonyl cyanide-p-trifluoromethoxyphenylhydrazone; and R/A, complex I/III inhibitors, rotenone/antimycin A. Oxygen consumption rate (OCR) in HOSE cells (**A**). Basal respiration: baseline OCR–OCR with R/A (**B**). ATP-linked respiration: baseline OCR–OCR with Oligo (**C**). Proton leak: OCR with Oligo–OCR with R/A (**D**). Maximal respiration: OCR with FCCP–OCR with R/A (**E**). Spare respiratory capacity: OCR with FCCP–OCR with baseline (**F**). Non-mitochondrial respiration: OCR with R/A (**G**). Baseline OCR/ECAR: baseline OCR divided by baseline ECAR (**H**). All experiments were carried out by using fresh preparations (n = 5–6). Data are shown as the means ± the standard error of the mean (SEM). One-way ANOVA followed by Tukey’s HSD (Honestly Significant Difference) post hoc analysis. FABP4i: inhibition of FABP4; FABP5i: inhibition of FABP5; PPARγi: inhibition of PPARγ.

**Figure 5 ijms-26-04794-f005:**
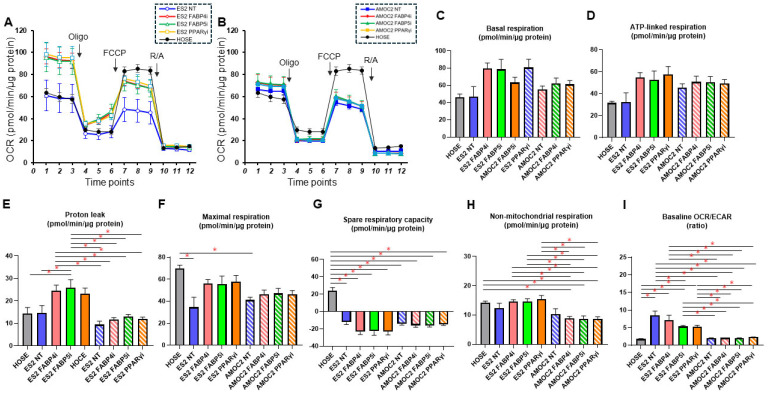
Comparison of metabolic functions of AMOC-2 cells and ES2 cells treated with lipid metabolism modulators with HOSE cells. Real-time cellular metabolic function analysis of HOSE, AMOC-2, and ES2 cells was performed to evaluate metabolic functions. OCR and ECAR were measured at the baseline and upon subsequent sequential supplementation with the following: Oligo, a complex V inhibitor, oligomycin; FCCP, a protonophore, carbonyl cyanide-p-trifluoromethoxyphenylhydrazone; and R/A, complex I/III inhibitors, rotenone/antimycin A. Oxygen consumption rate (OCR) in HOSE and AMOC-2 cells (**A**). OCR in HOSE and ES2 cells (**B**). Basal respiration: baseline OCR–OCR with R/A (**C**). ATP-linked respiration: baseline OCR–OCR with Oligo (**D**). Proton leak: OCR with Oligo–OCR with R/A (**E**). Maximal respiration: OCR with FCCP–OCR with R/A (**F**). Spare respiratory capacity: OCR with FCCP–OCR with baseline (**G**). Non-mitochondrial respiration: OCR with R/A (**H**). Baseline OCR/ECAR: baseline OCR divided by baseline ECAR (**I**). All experiments were carried out by using fresh preparations (n = 3–4). Data are shown as the means ± the standard error of the mean (SEM). * *p* < 0.05 by one-way ANOVA followed by Tukey’s HSD (Honestly Significant Difference) post hoc analysis. FABP4i: inhibition of FABP4; FABP5i: inhibition of FABP5; PPARγi: inhibition of PPARγ.

**Figure 6 ijms-26-04794-f006:**
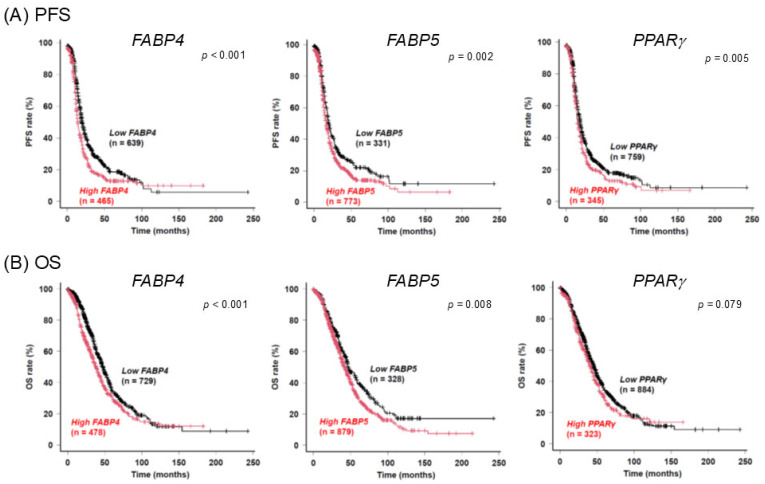
Associations of the expression levels of FABP4, FABP5, and PPARγ with PFS (**A**) and OS (**B**) in patients with serous ovarian carcinoma were evaluated using a Kaplan–Meier plotter. FABP4, fatty acid binding protein 4; FABP5, fatty acid binding protein 5; PPARγ, peroxisome proliferator-activated receptor gamma; PFS, progression-free survival; OS, overall survival.

**Figure 7 ijms-26-04794-f007:**
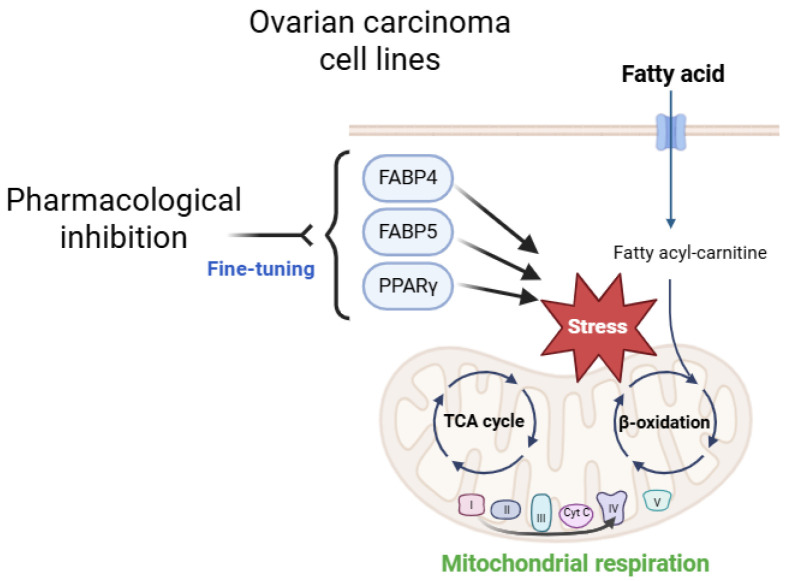
A simplified schematic figure illustrating the putative conceptual framework of the functional outcomes observed in this study.

**Figure 8 ijms-26-04794-f008:**
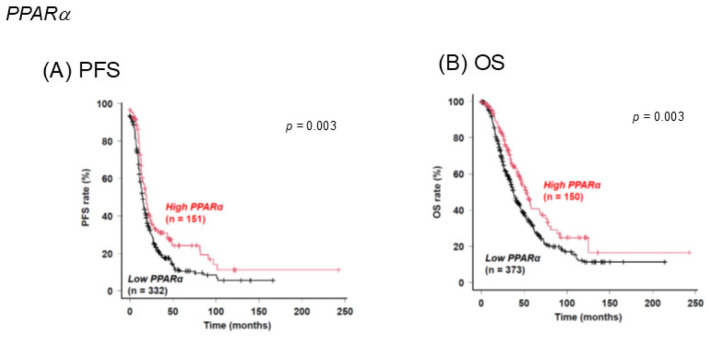
Associations of the expression levels of PPARα with PFS (**A**) and OS (**B**) in patients with serous ovarian carcinoma were evaluated using a Kaplan–Meier plotter. PPARα, peroxisome proliferator-activated receptor alpha; PFS, progression-free survival; OS, overall survival.

## Data Availability

The original contributions presented in this study are included in the article/Appendix A; further inquiries can be directed to the corresponding author.

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
