# Peer review of "Fatty Acid Metabolism Regulators Have Pivotal Roles in the Pathogenesis of Ovarian Carcinoma"

_ijms, 2025, doi:10.3390/ijms26104794_

Round 1
Reviewer 1 Report
Comments and Suggestions for Authors
The manuscript titled “Fatty acid metabolism regulators have pivotal roles in the pathogenesis of ovarian carcinoma” investigates the roles of fatty acid metabolism-related factors (FABP4, FABP5, and PPARγ) across various epithelial ovarian cancer (EOC) subtypes. The study uses 3D tumor spheroids, gene expression profiling, survival analyses from the TCGA database, and metabolic flux assessments via Seahorse XF analysis to elucidate the biological relevance of these regulators in ovarian cancer.
However, several critical concerns remain. First, the expression of metabolic regulators is evaluated solely at the mRNA level, without validation at the protein level. The absence of protein expression or localization data—such as western blotting or immunohistochemistry (IHC)—substantially limits the strength of the findings. It is strongly recommended that the authors provide protein quantification and localization data to enhance the credibility of their conclusions.
Furthermore, while the Seahorse analysis reveals differential responses to metabolic inhibitors, the mechanistic roles of FABP4, FABP5, and PPARγ in metabolic regulation remain inadequately defined. Additional experiments are recommended to establish a more direct functional link, such as fatty acid uptake assays, ROS detection, and evaluations of cell viability or motility following modulation of these metabolic pathways.
Finally, although the manuscript demonstrates a well-structured experimental design, robust data analysis, and strong potential for clinical translation, significant language and originality issues remain. According to the iThenticate report, the overall similarity index is 35%, well above the commonly accepted threshold for international journals (generally <20%). Most overlaps are concentrated in the background, methods, and portions of the results section. This likely stems from insufficient paraphrasing of existing literature, reliance on template language for describing standard procedures, and the use of public database outputs whose phrasing may resemble existing publications. While this degree of overlap may not constitute plagiarism, it detracts from the manuscript's perceived originality.
Therefore, I recommend Minor Revision, particularly addressing the lack of protein-level validation, strengthening mechanistic insights, and reducing textual similarity through thorough rewriting and improved paraphrasing.
Author Response
Dear Editor, Thank you very much for the constructive comments concerning our manuscript “Fatty acid metabolism regulators have pivotal roles in the pathogenesis of ovarian carcinoma”. We carefully checked all of the reviewers’ comments and prepared a revised version of our paper that takes these comments into account. The changes are listed below. Reviewer 1 comments The manuscript titled “Fatty acid metabolism regulators have pivotal roles in the pathogenesis of ovarian carcinoma” investigates the roles of fatty acid metabolism-related factors (FABP4, FABP5, and PPARγ) across various epithelial ovarian cancer (EOC) subtypes. The study uses 3D tumor spheroids, gene expression profiling, survival analyses from the TCGA database, and metabolic flux assessments via Seahorse XF analysis to elucidate the biological relevance of these regulators in ovarian cancer. 1. However, several critical concerns remain. First, the expression of metabolic regulators is evaluated solely at the mRNA level, without validation at the protein level. The absence of protein expression or localization data—such as western blotting or immunohistochemistry (IHC)—substantially limits the strength of the findings. It is strongly recommended that the authors provide protein quantification and localization data to enhance the credibility of their conclusions. Answer; We sincerely appreciate your excellent comment. As suggested, results of immunohistochemistry are included in Fig 3B. 2. Furthermore, while the Seahorse analysis reveals differential responses to metabolic inhibitors, the mechanistic roles of FABP4, FABP5, and PPARγ in metabolic regulation remain inadequately defined. Additional experiments are recommended to establish a more direct functional link, such as fatty acid uptake assays, ROS detection, and evaluations of cell viability or motility following modulation of these metabolic pathways. Answer; Thank you very much for your insightful comments. We agree that additional experiments including the uptake of various fatty acids and the detection of various types of ROS may provide further insight into the mechanism, but we believe that the current findings from the Seahorse metabolic analysis provide sufficient evidence to link FABP4, FABP5, and PPARγ to the metabolic regulation of ovarian cancer cells, which is necessary for the conclusions of this study. Specifically, we demonstrated that pharmacological inhibition of these regulatory factors restored the impaired maximal respiration observed in AMOC2 and ES2 cells, without affecting significant changes in basal respiration. The findings suggest that FABP4, FABP5, and PPARγ may influence energy metabolism by modulating electron transfer efficiency in the mitochondrial electron transport chain, the driving force of proton pumping, and/or the supply of NADH and FADH₂ through the TCA cycle and β-oxidation. These interpretations have been added to the 1st paragraph of the revised Discussion section. While we acknowledge that additional experiments including multi-omics analyses would be valuable for elucidating the underlying mechanisms, such studies are currently limited by constraints in time, effort, and resources as mentioned in the original version of Limitation section. To the best of our knowledge, this is the first study to investigate the involvement of these metabolic regulators in AMOC2 and ES2 cells, and we consider it appropriate to share these novel and meaningful findings with the scientific community at this stage. We appreciate your understanding of our decision to share these results with the scientific community while aiming to address the mechanistic questions in future work. 3. Finally, although the manuscript demonstrates a well-structured experimental design, robust data analysis, and strong potential for clinical translation, significant language and originality issues remain. According to the iThenticate report, the overall similarity index is 35%, well above the commonly accepted threshold for international journals (generally
Reviewer 2 Report
Comments and Suggestions for Authors
In "Fatty acid metabolism regulators have pivotal roles in the pathogenesis of ovarian carcinoma", Watanabe and co-authors describe a role for fatty acid (FA) metabolism regulator genes including FABP4/5, PPARa/g, and GPR40/41/43/120 in ovarian carcinoma. They also describe gene expression of the adherent junction genes ZO-1 and claudin 1. They then suggest a role for FA regulators in ovarian carcinoma.
This is an interesting subject, and the authors do a laudable job with gene expression analysis, clinical data analysis, and metabolic measurements. However, there are still questions marks on this study that prohibit me from accepting this study for publication:
- How did the authors decide to study these genes?
- The use of one cell line (HOSE) as "normal", and one each for OSC and OCCC is completely insufficient.
- What is the result for the GPR genes?
- Could the authors provide a figure on FA metabolism so the regulation and action of the genes studied can be correlated to the in vitro metabolism results?
Author Response
Dear Editor,
Thank you very much for the constructive comments concerning our manuscript “Fatty acid metabolism regulators have pivotal roles in the pathogenesis of ovarian carcinoma”. We carefully checked all of the reviewers’ comments and prepared a revised version of our paper that takes these comments into account. The changes are listed below.
Reviewer 1 comments
The manuscript titled “Fatty acid metabolism regulators have pivotal roles in the pathogenesis of ovarian carcinoma” investigates the roles of fatty acid metabolism-related factors (FABP4, FABP5, and PPARγ) across various epithelial ovarian cancer (EOC) subtypes. The study uses 3D tumor spheroids, gene expression profiling, survival analyses from the TCGA database, and metabolic flux assessments via Seahorse XF analysis to elucidate the biological relevance of these regulators in ovarian cancer.
- However, several critical concerns remain. First, the expression of metabolic regulators is evaluated solely at the mRNA level, without validation at the protein level. The absence of protein expression or localization data—such as western blotting or immunohistochemistry (IHC)—substantially limits the strength of the findings. It is strongly recommended that the authors provide protein quantification and localization data to enhance the credibility of their conclusions.
Answer; We sincerely appreciate your excellent comment. As suggested, results of immunohistochemistry are included in Fig 3B.
- Furthermore, while the Seahorse analysis reveals differential responses to metabolic inhibitors, the mechanistic roles of FABP4, FABP5, and PPARγ in metabolic regulation remain inadequately defined. Additional experiments are recommended to establish a more direct functional link, such as fatty acid uptake assays, ROS detection, and evaluations of cell viability or motility following modulation of these metabolic pathways.
Answer; Thank you very much for your insightful comments. We agree that additional experiments including the uptake of various fatty acids and the detection of various types of ROS may provide further insight into the mechanism, but we believe that the current findings from the Seahorse metabolic analysis provide sufficient evidence to link FABP4, FABP5, and PPARγ to the metabolic regulation of ovarian cancer cells, which is necessary for the conclusions of this study.
Specifically, we demonstrated that pharmacological inhibition of these regulatory factors restored the impaired maximal respiration observed in AMOC2 and ES2 cells, without affecting significant changes in basal respiration. The findings suggest that FABP4, FABP5, and PPARγ may influence energy metabolism by modulating electron transfer efficiency in the mitochondrial electron transport chain, the driving force of proton pumping, and/or the supply of NADH and FADH₂ through the TCA cycle and β-oxidation. These interpretations have been added to the 1st paragraph of the revised Discussion section.
While we acknowledge that additional experiments including multi-omics analyses would be valuable for elucidating the underlying mechanisms, such studies are currently limited by constraints in time, effort, and resources as mentioned in the original version of Limitation section. To the best of our knowledge, this is the first study to investigate the involvement of these metabolic regulators in AMOC2 and ES2 cells, and we consider it appropriate to share these novel and meaningful findings with the scientific community at this stage. We appreciate your understanding of our decision to share these results with the scientific community while aiming to address the mechanistic questions in future work.
- Finally, although the manuscript demonstrates a well-structured experimental design, robust data analysis, and strong potential for clinical translation, significant language and originality issues remain. According to the iThenticate report, the overall similarity index is 35%, well above the commonly accepted threshold for international journals (generally <20%). Most overlaps are concentrated in the background, methods, and portions of the results section. This likely stems from insufficient paraphrasing of existing literature, reliance on template language for describing standard procedures, and the use of public database outputs whose phrasing may resemble existing publications. While this degree of overlap may not constitute plagiarism, it detracts from the manuscript's perceived originality.
Answer; We sincerely appreciate your excellent comment. As pointed out, overall similarity in the present manuscript was reduced. Changes are shown as blue-hilighted.
- Therefore, I recommend Minor Revision, particularly addressing the lack of protein-level validation, strengthening mechanistic insights, and reducing textual similarity through thorough rewriting and improved paraphrasing. For PPAR alpha activation, it has been reported to change the expression of ocular FABP4. Screening data from the liver, retina, and RPE-Choroid showed that FABP4 expression could be increased by a PPAR alpha activator (7717/peerj.14611). There is a discrepancy in vitro and in vivo. Depending on the time point and the dose of the activator, the trend could be changed in vitro. This should be at least discussed.
Answer; We sincerely appreciate your excellent comment and idea. As suggested, protein level validation by using immunocytochemistry is included, and through thorough rewriting and paraphrasing are revised to strengthen mechanistic insights and to reduce textual similarity. In addition, as according your excellent idea, discussion related to previous results related to PPAR alpha activator alter the expression of ocular FABP4 by Screening data from the liver, retina, and RPE-Choroid (DOI10.7717/peerj.14611) is included in last part of 3rd paragraph of Discussion: ‘Furthermore, a previous screening study using the liver, retina, and retinal pigment epithelium (RPE)-choroid showed that PPARa activation increased the expression of ocular FABP4 [58] suggesting that PPARa may also be involved in the pathogenesis of OC through some unidentified linkage with FABP4. In support of this idea, levels of PPARa expression in HOSE cells were slightly lower than those in AMOC-2 and ES2 cells, and SOC patients with lower expression level of PPARa (Fig.3) and a Kaplan-Meier plotter using an online database showed significantly shorter PFS and overall survival (OS) than those for SOC patients with their higher expression level of PPARa (Fig 8).’.
Reviewer 2 comments
In "Fatty acid metabolism regulators have pivotal roles in the pathogenesis of ovarian carcinoma", Watanabe and co-authors describe a role for fatty acid (FA) metabolism regulator genes including FABP4/5, PPARa/g, and GPR40/41/43/120 in ovarian carcinoma. They also describe gene expression of the adherent junction genes ZO-1 and claudin 1. They then suggest a role for FA regulators in ovarian carcinoma.
This is an interesting subject, and the authors do a laudable job with gene expression analysis, clinical data analysis, and metabolic measurements. However, there are still questions marks on this study that prohibit me from accepting this study for publication:
- How did the authors decide to study these genes?
Answer; We sincerely appreciate your excellent comment. In terms of decision of study genes related to ovarian carcinoma, previous studies have suggested that among various FA metabolism regulators, fatty binding acid protein 4 (FABP4) [47][48] and peroxisome proliferator-activated receptors (PPARs) [51, 52][53] are expressed in the ovarian tissues and are involved in the pathogenesis of OC. In addition, it was shown that FABP5 also identified as a pivotal factor regulating the pathogenesis of various malignant tumors [56]. Thus, based on these, we decided to study these factors. In contrast, GPRs have not been studied in terms of OC pathogenesis and in fact, there were no expression of them in three cell lines derived from ovarian tissues in the present study. Taken together, results of GPRs are removed and above information is included in 2nd paragraph of Introduction: ‘Premature menstruation, nulligravida, advanced maternal age and late menopause have been shown to be risk factors for EOC progression [10]. Obesity is also recognized as a risk factor for the development and progression of EOC and other malignant tumors [11-13]. It has been shown that obesity is negatively correlated with the survival of patients with EOC by affecting their metastasis [14], and a high visceral fat-to-muscle ratio is known as an independent factor for predicting deteriorated overall survival (OS) in patients with EOC [15]. These findings collectively suggest that fatty acid (FA) metabolism factors such as adipokines may be involved in the pathogenesis of EOC, and, in fact, within tumor surrounding microenvironments, which have crucial roles in tumorigenesis tumor progression, and metastasis [16-19], several adipokines in addition to other factors including monocyte chemotactic protein-1 (MCP-1), interleukin 6 (IL-6), interleukin 8 (IL-8), tissue inhibitor of metalloproteinase-1 (TIMP-1) vascular endothelial growth factor-A (VEGF-A) and matrix metalloproteoinase-2 (MMP-2) have been detected [20-22]. In fact, among various FA metabolism regulators, previous studies have suggested that fatty binding acid protein 4 (FABP4) [23, 24] and peroxisome proliferator-activated receptors (PPARs) [25-27] are expressed in the ovarian tissues and are involved in the pathogenesis of OC. In addition, it was shown that FABP5 also identified as a pivotal factor regulating the pathogenesis of various malignant tumors [28]. However, at the time of this writing, the contribution of these FA metabolism regulators to the pathogenesis of EOC has not been intensively investigated.’.
- The use of one cell line (HOSE) as "normal", and one each for OSC and OCCC is completely insufficient.
Answer; We sincerely appreciate your excellent comment. As pointed out, I agree that only three cell lines are insufficient to propose a new concept. However, I believe that comparison of various biological aspects related to fatty acid regulators among three distinct cell types derived from normal and malignant ovarian tissues still provide quite new information as a pilot study in this research field. Therefore, this information is included in the study limitation in Discussion section: ‘Nevertheless, as study limitations, the following issues need to be investigated. Firstly, only three distinct cell lines derived from healthy and two cell lines originating from OSC and OCCC were used in the present study. Secondly, underlying mechanisms inducing different metabolic phenotypes between AMOC-2 cells (glycolysis dominant) and ES2 cells (mitochondrial respiration dominant) assessed by a Seahorse Bioanalyzer remain to be elucidated. Since it has been shown that more than half of OCCC cases are associated with inactivating mutations of ARID1A, a component of the SWI/SNF nucleosome remodeling complex [64], and that ARID1A-deficient cells upregulate OXPHOS [65], ARID1A mutation may be involved. However, another study showed that although OCR indices of the OCCC cell line ES2 with ARID1A wildtype and another OCCC cell line TOV-21G cells with ARID1A mutation were substantially higher than those of OACC cell lines, levels of OCR indices of ES2 cells and TOV-21G cells were comparable, suggesting that the difference in metabolic phenotypes between ES2 cells and AMOC-2 cells is not related to ARID1A mutation. Thirdly, possible pathophysiological roles of FABPs and/or PPARs in OC cells have not been identified despite the fact that their specific inhibitors substantially altered cellular metabolic functions. Therefore, further studies to reveal those unidentified issues in conjugation with additional investigations using additional various OC cell lines to find new key molecules related to FA metabolism regulators and the correlation between expression of FA metabolism regulators and various clinical parameters, such histological grade and FIGO staging in patients with OC will be our next projects.’.
- What is the result for the GPR genes?
Answer; We sincerely appreciate your excellent comment. In terms of GPR genes, mRNA expression of those genes was not observed in three cell lines and thus, further study related to these genes were not performed. In addition, as according to above comment 1, there have been much less studies related to GPR genes compared to FABLs and PPARs in terms of OC research field as fur as I survey. Therefore, GPR genes’ related issues are removed.
- Could the authors provide a figure on FA metabolism so the regulation and action of the genes studied can be correlated to the in vitro metabolism results?
Answer; We sincerely appreciate your excellent suggestion. We totally agree that a visual representation of FA metabolism could help understanding the regulatory roles of FABP4, FABP5, and PPARγ in relation to our in vitro metabolic analysis. Although we are convinced that the results contain novel findings, we did not identify the detailed underlying mechanisms. Taken this limitation into account, we have now made a simplified schematic figure (Revised Figure 7). This Figure is intended to serve as a conceptual framework to support our observed functional outcomes, rather than a comprehensive map of fatty acid metabolism. We hope that the schema will help the readers’ understanding.
Reviewer 3
Watanabe and colleagues present an interesting work evaluating some fatty acids regulators and adhesion spheroid-related molecules in normal and two different ovarian cancer cell lines. Some improvements are suggested below to increase the quality of the paper:
- Lines 56-61: The authors could mention at this point a recent related study that shows how some extracellular signaling molecules like TIMP-1, VEGF-A, and MMP-2 from adipose-derived stem cells lead to ovarian cancer progression (please see https://doi.org/10.3390/cells14050374).
Answer; We sincerely appreciate your excellent comment. As pointed out, this phrase is revised to including suggested information: ‘These findings collectively suggest that fatty acid (FA) metabolism factors such as adipokines may be involved in the pathogenesis of EOC, and, in fact, within tumor surrounding microenvironments, which have crucial roles in tumorigenesis tumor progression, and metastasis [16-19], several adipokines in addition to other factors including monocyte chemotactic protein-1 (MCP-1), interleukin 6 (IL-6), interleukin 8 (IL-8), tissue inhibitor of metalloproteinase-1 (TIMP-1) vascular endothelial growth factor-A (VEGF-A) and matrix metalloproteoinase-2 (MMP-2) have been detected [20-22]. In fact, among various FA metabolism regulators, previous studies have suggested that fatty binding acid protein 4 (FABP4) [23, 24] and peroxisome proliferator-activated receptors (PPARs) [25-27] are expressed in the ovarian tissues and are involved in the pathogenesis of OC. In addition, it was shown that FABP5 also identified as a pivotal factor regulating the pathogenesis of various malignant tumors [28]. However, at the time of this writing, the contribution of these FA metabolism regulators to the pathogenesis of EOC has not been intensively investigated.’.
- Lines 61 and 245: The authors use the sentence "as of this writing," which sounds incomplete. I suggest changing it to "at the time of this writing."
Answer; We sincerely appreciate your excellent comment. As pointed out, this phrase is changed.
- Lines 95-102: This information should be removed to the material and methods section.
Answer; We sincerely appreciate your excellent comment. As pointed out, these are moved to method section and corresponding result: ‘In previous studies, it was shown that non-globe shaped tumor cell-derived spheroids were produced by using a hanging drop culture method, in which only gravity force and buoyant force are required for generating spheroids [24-27, 30-34], suggesting that the appearance of spheroids may be a key index for some important biological aspects of tumorgenesis [35]. Initially, to compare the configuration of spheroids generated non-cancerous HOSE cells, AMOC-2 cells established from OSC and ES2, cells established from OCCC cells, downward and lateral images of spheroids were observed by phase contrast microscope and a microscope monitoring camera, respectively as described in our recent report [36].’ and method: ‘To study the morphological aspects of the spheroids, each spheroid was observed downwardly and laterally by using a phase contrast microscopy as our previous reports [26]. Briefly, the horizontal configurations of spheroids that were dropped into a petri dish from wells of a 3D drop culture plate were observed by a phase contrast microscope. The spheroids were then transferred onto a microplate equipped with MicroSquisher (CellScale, Waterloo, ON, Canada) for observation of lateral images of spheroids observed by a microscope monitoring camera were obtained. For measurement of the horizontal diameter and lateral height of 3D spheroids were processed by using Image-J software version 1.51n (National Institutes of Health, Bethesda, MD) as described in our recent report [36].’ are revised.
- Line 149: What patients? OC patients? Please include this information also in this point of the text.
Answer; We sincerely appreciate your excellent comment. As pointed out, this is changed to SOC patients.
- Figures 4 and 5 are of poor quality. Please increase their size and provide high-quality images in the manuscript.
Answer; We sincerely appreciate your excellent comment. As suggested, quality of these figures is improved as possible as we can.
- Line 295: Typo error in "2D". The subsection must be in italics.
Answer; We sincerely appreciate your excellent comment. As pointed out, this type error is fixed.
- Section 4.1.: The authors should include the methodological detail for the obtention of the spheroids despite the fact that the method was previously described in another work.
Answer; We sincerely appreciate your excellent comment. As pointed out, brief method for spheroid generation is included: ‘The 2D and 3D cell cultures of these cell lines were processed during a period of 7 days as we reported previously [67, 68]. Briefly, as for spheroid generation, 2D cultured cell pellet was collected and resuspended in the above DMEM culture medium supplemented with methylcellulose (Methocel A4M) in which cell density was adjusted at 20,000 cells / 28 ml. A 28-ml aliquot of the cell suspension was subjected in each well of a hanging drop culture plate (# HDP1385, Sigma-Aldrich). Spheroid culture was processed by daily medium changing of half of the medium in each well.’.
- Line 304: "tow2d"? Please revise.
Answer; We sincerely appreciate your excellent comment. As pointed out, this typo error is corrected.
- Section 4.2.: Why did the authors not include any morphometrical analysis of the spheroids?
Answer; We sincerely appreciate your excellent comment. As suggested, for morphometrical analysis, results of horizontal diameter and lateral height are included in Fig. 1.
- Line 328: "Twenty thousand cells." This information is usually shown in numerals elevated to the potency of 10.
Answer; We sincerely appreciate your excellent comment. As pointed out, this is corrected to 104 cells.
Reviewer 4 comments
The study by Watanabe et al. deals with the role of fat acid metabolism in the pathogenesis of ovarian carcinoma. By using multiparametric analyses ((1) analysis of the physical properties of spheroids, 2) qPCR analysis, 3) multiomic pancancer analysis and 4) cellular metabolic analysis). they first found different globe-shaped spheroids of non tumoral HOSE cells, and non-globe-shaped in both cancer ES2 an AMOC-2 spheroids. Furthermore, the Authors observed an increase of gene expression levels of FABP4 and FABP5 in ES2 cells and PPARγ in both cancer cells as compared to non tumoral HOSE cells. Subsequently, they did comparative analyses of associations of the expression levels of FABP4, FABP5 and PPARγ with PFS and OS in patients with serous ovarian carcinoma. At the end, the Authors investigated the metabolic functions (oxygen consumption rate, basal respiration, ATP-linked respiration and proton leak in HOSE, AMOC2 and ES2 cells. The data are interesting and go along several other lines of evidence for lipid metabolism in ovarian cancer. However, I do think that the data is somewhat over-interpreted and some important results are lacking so one can analyse the biological relevance of the data presented.
Detailed comments regarding this matter are outlined below:
- Exploring the correlation between gene expression of FA metabolism regulators and various clinical parameters, such histological grade and FIGO staging in patients with ovarian cancer would provide valuable insights.
- Regarding the selection of cell lines, data from different OC cell lines should be included. This would offer a more comprehensive context for understanding the effects of FA metabolism in OC.
Answers for comments #1 and #2: Thank you very much for your insightful comments. AMOC-2, an ovarian serous adenocarcinoma (OSC) cell line, that were generously supplied by Dr. Hiromitsu Yabushita, Department of Obstetrics and Gynecology, Aichi Medical college [Yabushita, H. Acta Obstet Gynaecol Jpn 1989, 41, 888-94.]. ES2, an ovarian clear cell carcinoma (OCCC) cell line, that were purchased from American Type Culture Collection (ATCC, Manassas, VA, USA). ES-2 is a cell line exhibiting fibroblast-like morphology that was isolated from the ovary of a Black, 47-year-old, female human with clear cell carcinoma. Therefore, for collection of cell lines in the present study, two major common OC cell types related cell lines were used. In ovarian cancer, it has been well known that dysregulated FA metabolism is associated with aggressive behavior, chemoresistance, and poor prognosis including high-grade serous carcinoma (HGSC), the most common and aggressive subtype of epithelial ovarian cancer (EOC), and clear cell carcinoma. In fact, it is also shown that lipogenic genes like FASN and SCD1 are positively correlated with higher histological grade in EOC, and there is a clear correlation between high expression of lipogenic FA metabolism regulators and worse clinical parameters. As suggested, I totally agree that exploring the correlation between gene expression of FA metabolism regulators and various clinical parameters, such histological grade and FIGO staging in patients with ovarian cancer would provide valuable insights, and therefore, this information id included in the last sentence of study limitation in the Discussion: ‘Therefore, further studies to reveal those unidentified issues in conjugation with additional investigations using additional various OC cell lines to find new key molecules related to FA metabolism regulators and the correlation between expression of FA metabolism regulators and various clinical parameters, such histological grade and FIGO staging in patients with OC will be our next projects.’.
- The impact of FABP and PPARγ inhibitors on cell biology of spheroids should be further substantiated by analysing the cell proliferation and stem markers.
Answer; We sincerely appreciate your excellent comment. In terms of cell proliferation, our previous studies in which total cell numbers in single cell derived 3D spheroids were evaluated have shown that, in contrast to 2D planar cell culture, cell proliferation was not observed in the 3D spheroid culture (PMID: 34588570, PMID: 37511214, PMID: 39329734). Therefore, we suggested that single cell derived 3D spheroid may develop like an immature organ instead of simple cell proliferation observed in conventional 2D planar cell culture. Furthermore, in our previous study using RNA sequencing analysis, we compared gene expression profiles related to adipogenic proliferation in 3T3-L1 mouse preadipocytes between 2D and 3D cell culture system (PMID: 35053416, PMID: 37867843, PMID: 393297349). Quite interestingly, differentially expressed genes (DEGs) upon adipogenesis, although various abundant genes including cell growth, stem cell markers, and others in addition to less abundant adipogenesis-related factors were detected in 2D planar cell culture, most of DEGs were adipogenesis-related genes and other less abundant genes including mitochondrial related genes in 3D spheroid culture (PMID: 37867843, PMID: 393297349). These our previous observation supported FABPs and PPARs inhibitors affected to cell metabolism but not cell proliferation, and therefore, I assume that analysis of cellular metabolisms would be important rather than cell proliferation and stem markers.
- The last section of the results presents the key experiments that led to the main message of this study, i.e. the cell metabolism following FABP and PPARγ inhibitors. However, one key experiment is missing and should be performed with non tumoral HOSE cells and FABP/PPARγ inhibitors, since the gene expression FABP4 and FABP5 are comparable to AMOC2 and ES2 cells. Regarding the use of use of spheroid microplate assay workflow with Seahorse should better described. Did the authors use a protocol suitable for 3D cells rather than 2D cells?
Answer; Thank you for these excellent suggestions. In accordaince to the former suggestion, We performed additional experiments using 2D-cultured HOSE cells with inhibitors of FABPs and PPARγ (Revised Figure 5). Interestingly, unlike in tumor cells, we did not observe enhancement of mitochondrial respiration by these inhibitors. These findings support the notion that FABPs and PPARγ have effects on mitochondrial function specifically in tumor cells. Regarding the metabolic assays in 3D spheroids, we do have the protocol available (PMID: 35740359, PMID: 37867843). However, the 3D spheroids derived from “cancer cells” were fragile, making it difficult to apply an equal amount to each well. We hope that the reviewers will understand these technical limitations.

Reviewer 3 Report
Comments and Suggestions for Authors
Watanabe and colleagues present an interesting work evaluating some fatty acids regulators and adhesion spheroid-related molecules in normal and two different ovarian cancer cell lines. Some improvements are suggested below to increase the quality of the paper:
A) Lines 56-61: The authors could mention at this point a recent related study that shows how some extracellular signaling molecules like TIMP-1, VEGF-A, and MMP-2 from adipose-derived stem cells lead to ovarian cancer progression (please see https://doi.org/10.3390/cells14050374).
B) Lines 61 and 245: The authors use the sentence "as of this writing," which sounds incomplete. I suggest changing it to "at the time of this writing."
C) Lines 95-102: This information should be removed to the material and methods section.
D) Line 149: What patients? OC patients? Please include this information also in this point of the text.
E) Figures 4 and 5 are of poor quality. Please increase their size and provide high-quality images in the manuscript.
F) Line 295: Typo error in "2D". The subsection must be in italics.
G) Section 4.1.: The authors should include the methodological detail for the obtention of the spheroids despite the fact that the method was previously described in another work.
H) Line 304: "tow2d"? Please revise.
I) Section 4.2.: Why did the authors not include any morphometrical analysis of the spheroids?
J) Line 328: "Twenty thousand cells." This information is usually shown in numerals elevated to the potency of 10.
Author Response

(The authors gave the same response as above.)

Reviewer 4 Report
Comments and Suggestions for Authors
The study by Watanabe et al. deals with the role of fat acid metabolism in the pathogenesis of ovarian carcinoma. By using multiparametric analyses ((1) analysis of the physical properties of spheroids, 2) qPCR analysis, 3) multiomic pancancer analysis and 4) cellular metabolic analysis). they first found different globe-shaped spheroids of non tumoral HOSE cells, and non-globe-shaped in both cancer ES2 an AMOC-2 spheroids. Furthermore, the Authors observed an increase of gene expression levels of FABP4 and FABP5 in ES2 cells and PPARγ in both cancer cells as compared to non tumoral HOSE cells.. Subsequently, they did comparative analyses of associations of the expression levels of FABP4, FABP5 and PPARγ with PFS and OS in patients with serous ovarian carcinoma.. At the end, the Authors investigated the metabolic functions (oxygen consumption rate, basal respiration, ATP-linked respiration and proton leak in HOSE, AMOC2 and ES2 cells. The data are interesting and go along several other lines of evidence for lipid metabolism in ovarian cancer. However, I do think that the data is somewhat over-interpreted and some important results are lacking so one can analyse the biological relevance of the data presented.
Detailed comments regarding this matter are outlined below:
Exploring the correlation between gene expression of FA metabolism regulators and various clinical parameters, such histological grade and FIGO staging in patients with ovarian cancer would provide valuable insights.
Regarding the selection of cell lines, data from different OC cell lines should be included. This would offer a more comprehensive context for understanding the effects of FA metabolism in OC .
The impact of FABP and PPARγ inhibitors on cell biology of spheroids should be further substantiated by analysing the cell proliferation and stem markers.
The last section of the results presents the key experiments that led to the main message of this study, i.e. the cell metabolism following FABP and PPARγ inhibitors. However, one key experiment is missing and should be performed with non tumoral HOSE cells and FABP/PPARγ inhibitors, since the gene expression FABP4 and FABP5 are comparable to AMOC2 and ES2 cells. Regarding the use of use of spheroid microplate assay workflow with Seahorse should better described. Did the authors use a protocol suitable for 3D cells rather than 2D cells?
Author Response

(The authors gave the same response as above.)

Round 2
Reviewer 2 Report
Comments and Suggestions for Authors
The authors have addressed some of my concerns, but do not feel obliged to extend their in vitro experiments using more cell lines.
Althought the body of work in this manuscript might be convincing as a whole, I still seriously object to experiments using just 1 cell line per cancer type.
Author Response
Dear Editor,
Thank you very much for the constructive comments concerning our manuscript “Fatty acid metabolism regulators have pivotal roles in the pathogenesis of ovarian carcinoma”. We carefully checked all of the reviewers’ comments and prepared a revised version of our paper that takes these comments into account. The changes are listed below.
Reviewer 2 comments
The authors have addressed some of my concerns, but do not feel obliged to extend their in vitro experiments using more cell lines. Althought the body of work in this manuscript might be convincing as a whole, I still seriously object to experiments using just 1 cell line per cancer type.
Answer; We sincerely appreciate your critical comment. Again, as pointed out, I totally agree that only three cell lines are insufficient to propose a new concept. We know that it is generally thought that more than one cell lines should be required for this kind of research field, and therefore, we used two epithelial ovarian carcinoma (EOC) cell lines, AMOC2 and ES2 in the present study, and our description may lead misunderstanding that only one cell line per each pathological types of OC. We apologized that overstatement that main purpose of the current study may be to elucidate different aspects between serous adenocarcinoma and clear cell carcinoma of OC, but our real main study purpose was to elucidate possible contribution of FA metabolism regulators in the pathogenesis of EOC. In fact, previous observations have shown that biological aspects especially gene expression profiles were significantly diverse among EOC even though in the same pathological types (ES2 cell: Genome Med. 2021 Sep 1;13(1):140. Cells. 2020 Nov 3;9(11):2408., AMOC-2 cell: Gynecol Oncol. 2000 Nov;79(2):256-63., Jpn J Cancer Res. 2002 Jun;93(6):644–651.). Therefore, we minimally selected two EOC cell lines ES2 and AMOC2 which are originated from ovarian serous adenocarcinoma and ovarian clear cell carcinoma, respectively, among various EOC cell lines rather than using 3-5 cell lines with same pathological background as a pilot study. In addition, current results that 3D spheroid appearance of AMOC2 and ES2 derived spheroid were non-globe shape as compared to globe shaped spheroid derived from non-cancerous HOSE cells are quite rationale to our recent observation that non-cancerous cells derived spheroid induce globed shaped spheroid but cancerous cells derived spheroid induce non-globe shaped spheroid beyond different tissues and species (reviewed in Cells. 2024 Sep 14;13(18):1549). Therefore, we are sure that two EOC cell lines and HOSE are indeed ovarian epithelial originated malignant tumors and non-malignant cells, and observed differences in the biological aspects between two EOC cell lines and HOSE suggested that FA metabolism regulators may indeed play some important roles in the pathogenesis of EOC even in taking consideration of diverse biological aspects among EOC cell lines. Therefore, to avoid these misleading, whole manuscript is revised (changes are highlighted).
Reviewer 4 comment
In the revised version of the manuscript, the authors conducted new experiments as suggested by the referee and included the results. These revisions have enhanced the quality and significance of the data. I believe their work could constitute an original contribution to the current state of research in this field.
Answer; We sincerely appreciate your heartful and encourage comment for our current study.

Reviewer 4 Report
Comments and Suggestions for Authors
In the revised version of the manuscript, the authors conducted new experiments as suggested by the referee and included the results. These revisions have enhanced the quality and significance of the data. I believe their work could constitute an original contribution to the current state of research in this field.
Author Response

(The authors gave the same response as above.)

Round 3
Reviewer 2 Report
Comments and Suggestions for Authors
The authors have answered my concern on the use of an inadequate amount of cell lines for the study. They have explained their rationale and reasoning for the use of these cell lines.
I still think the set-up of the study is not sufficient to support all conclusions drawn by the authors, but I will not object to publication.